# Microbial Enzyme Biotechnology to Reach Plastic Waste Circularity: Current Status, Problems and Perspectives

**DOI:** 10.3390/ijms24043877

**Published:** 2023-02-15

**Authors:** Marco Orlando, Gianluca Molla, Pietro Castellani, Valentina Pirillo, Vincenzo Torretta, Navarro Ferronato

**Affiliations:** 1Department of Biotechnology and Life Sciences, University of Insubria, Via Dunant, 21100 Varese, Italy; 2Department of Theoretical and Applied Sciences (DiSTA), University of Insubria, Via G.B. Vico 46, 21100 Varese, Italy

**Keywords:** circular bioeconomy, biotechnology, solid waste management, biodegradation, synthetic plastic

## Abstract

The accumulation of synthetic plastic waste in the environment has become a global concern. Microbial enzymes (purified or as whole-cell biocatalysts) represent emerging biotechnological tools for waste circularity; they can depolymerize materials into reusable building blocks, but their contribution must be considered within the context of present waste management practices. This review reports on the prospective of biotechnological tools for plastic bio-recycling within the framework of plastic waste management in Europe. Available biotechnology tools can support polyethylene terephthalate (PET) recycling. However, PET represents only ≈7% of unrecycled plastic waste. Polyurethanes, the principal unrecycled waste fraction, together with other thermosets and more recalcitrant thermoplastics (e.g., polyolefins) are the next plausible target for enzyme-based depolymerization, even if this process is currently effective only on ideal polyester-based polymers. To extend the contribution of biotechnology to plastic circularity, optimization of collection and sorting systems should be considered to feed chemoenzymatic technologies for the treatment of more recalcitrant and mixed polymers. In addition, new bio-based technologies with a lower environmental impact in comparison with the present approaches should be developed to depolymerize (available or new) plastic materials, that should be designed for the required durability and for being susceptible to the action of enzymes.

## 1. Introduction

Plastic waste management is a topic of concern at a global level [1,2,3]. It has been estimated that about 53 kilo-tons per year (kt y^−1^) of plastic waste will be released into the environment by 2030 [4]. This is due to inappropriate behaviors, lack of collection systems, or leakages from transportation [5]. Environmental plastic pollution is not only a local issue of contaminated soils and water bodies, but it is pervasive in worldwide ecosystems, also affecting polar regions and areas with no apparent human activity [6,7,8]. This is because plastic materials permeate through globally interconnected aquatic waste streams and are fragmented and chemically modified by abiotic processes into pieces of various sizes, from macroscopic to microscopic (nano- and microplastics) [9,10,11]. It has been recently reported that nano- and microplastics have entered biological food chains, tissues, and human blood [2,12,13]; such plastic particles are harmful for ecosystems and human health [14,15,16,17] and can interact with (and act as carriers of) toxic compounds and pathogens [10,18,19]. Therefore, new urgent regulations are expected to be released in the next few years to reduce plastic leakages into the environment. This will result in a higher amount of plastic waste to be managed and valorized. In such a scenario, circular-economy-based production systems designed to convert waste to new resources, instead of useless compounds to get rid of, will be favored [20,21,22].

In the plastic waste management field, circular economy principles are achieved by reducing, reusing, and recycling while simultaneously increasing the lifespan of materials and products. Among such activities, recycling is implemented mainly in high-tech infrastructure demanding considerable resource, time, and financial investments. In recent years, plastic recycling has been characterized by a decrease in costs [23,24] and an increase in efficiency; mechanical recycling is the most utilized approach [25]. However, it is a downcycling process that is suitable to recycle few plastic fractions and leads to products of lower quality than those from feedstock materials [26,27]. Furthermore, the market for recyclable materials has oscillating prices [28]. Therefore, future approaches aimed at potentiating plastic waste circularity and its revenues will benefit from more efficient and greener recycling technologies.

The biotechnology sector (mainly based on microbial enzyme applications) has recently emerged as a source of opportunities for waste circularity within the circular (bio)economy framework, as also stated by the European Union (EU) [29]. Enzymatic bioprocesses can convert biomass- or fossil-derived low-value waste into new and marketable products of comparable or superior quality in comparison with virgin materials, decreasing the CO_2_ footprints of newly extracted fossil feedstocks. Several reviews of available biotechnological tools in plastic waste circularity have been published in recent years (Table 1 for some recent examples), but none reported a combined view of the potential of biotechnological approaches in the context of current plastic waste treatments. In this review, we report an updated list of biotechnological tools for the depolymerization of plastics made with traditional petroleum-based non-renewable polymers (e.g., PET, polyethylene (PE), polypropylene (PP), and polystyrene (PS)), which represent most of the plastic produced worldwide and need centuries to completely degrade if spread in natural environments [30]. We focused on relative performances, practical opportunities, and limitations within the framework of current plastic waste flows and traditional recycling strategies. In doing so, we attempted to answer the following research question: “What is the contribution of biotechnology in plastic waste recycling?”. Finally, we provided the most promising directions to potentiate the contribution of biotech approaches in plastic waste valorization within the context of the circular (bio)economy.

## 2. Plastics: From Waste Generation to Recycling

### 2.1. Plastic Classification

In 2019, global plastic production reached 368,000 kt y^−1^ [44]. Thermoplastics and thermosets are the two categories of polymers into which plastics are divided [45]. Thermoplastics are composed of linear polymeric chains bridged by non-covalent hydrophobic interactions [46], allowing for their melting above a certain temperature and their hardening when cooled. This means that thermoplastics can be repeatedly reheated, reshaped, and cooled (with possible loss of performance) [47]. There are three types of thermoplastic polymers [48]: (i) In crystalline thermoplastic polymers, the regular arrangement of chains makes them translucent. They have more mechanical impact resistance compared to other polymers. Examples are polyethylene (low-density polyethylene (LDPE) and high-density polyethylene (HDPE)) and PP. (ii) In amorphous thermoplastic polymers, random chain arrangement usually provides transparency. Polyvinylchloride (PVC), polycarbonate (PC), polymethylmethacrylate (PMMA), acrylonitrile butadiene styrene (ABS), and PS are typical examples. (iii) Semicrystalline thermoplastic polymers have both crystalline and amorphous regions. Some examples are PET, polyamides (PAs), and polyester polybutylene terephthalate (PBT).

Thermosets are composed of branched polymeric chains that form a crosslinked three-dimensional network when heated. As these cross links are made up of covalent bonds, after their formation, these plastics cannot be recycled by melting and reforming [49]. In compliance with EU plastic demand distribution by resin type (Table 2), thermoplastics (about 85%) dominate over thermosets (around 15%), with crystalline thermoplastics being the most requested (up to 50%). Thermosets are composed of 51.3% PURs (5445 kt) and by other thermosets, such as phenolic resins, epoxide resins, melamine resins, and urea resins, among others.

### 2.2. EU Plastic Waste Flows

About 66,786 kt of plastic materials were produced in 2016 in the EU (Figure 1a): 80% (53,264 kt) with thermoplastic polymers and 18% (12,137 kt) with thermoset polymers. The remaining 2% (1385 kt) includes fibers made both with thermoset and thermoplastic polymers. In 2016, the amount of consumed plastic in products was 73,481 kt. Packaging products are the most consumed (19,461 kt, up to 26% of the total), followed by a fraction (18,101 kt, 25% of the total) that includes parts of furniture; the manufacture of plastic plates, sheets, tubes, profiles, and bleached paper; the manufacture of fabricated metal products; and other manufacturing (Figure 1b). Construction and transport products account for 12,297 kt and 10,545 kt, respectively, followed by electrical and electronic equipment (EEE) (6040 kt), textiles (4521 kt), health care (1112 kt), and paints and varnishes (1404 kt).

Clustering of consumed plastics in these categories of products is useful, reflecting the average time after which they move into the waste stream, forming so-called post-consumer plastic waste (PCPW). Plastic construction materials and transport-related products are assumed to remain in use in the sociotechnological system for between 30 and 50 years [50], contributing the most to form the so-called “stock” fraction, which amounted to ≈50% (37,696 kt) of all plastic products in 2016 [51].

**Figure 1 ijms-24-03877-f001:**
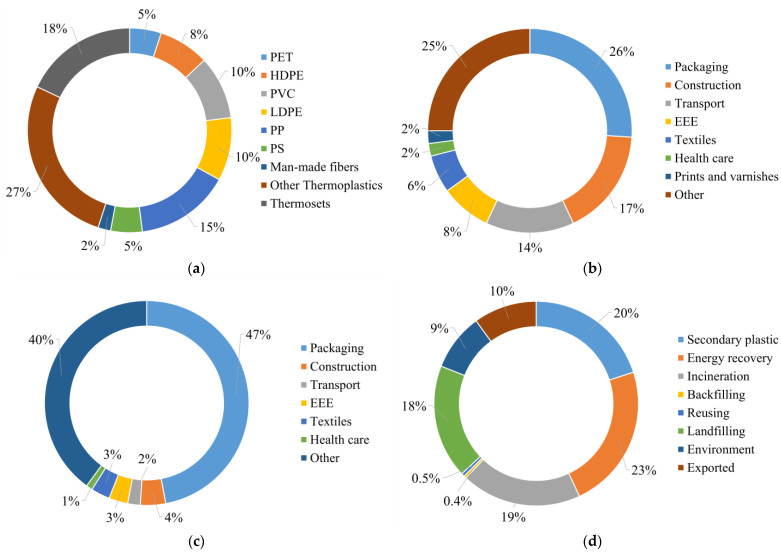
Plastics flows in the EU (2016): (**a**) production; (**b**) product consumption; (**c**) waste generation; (**d**) destination of waste. Elaborated from data presented by Hsu et al. [51].

PCPW (36,592 kt in 2016) can be divided into seven different flows (Figure 1c): packaging (16,122 kt), construction materials (1537 kt), transport waste (777 kt), EEE (984 kt), textiles (1101 kt), healthcare waste (233 kt), and “other” (13,601 kt). Plastic waste from manufacturing processes and from packaging products (in use for one year or less) contributes the most (≈50%) to PCPW, followed by the “other” fraction (≈35%), a heterogeneous mix of waste streams collected from various economic activities (markets, street cleaning, and municipal activities). Therefore, the “other” fraction is difficult to be correctly recognized, separated, and managed immediately after consumption. Adding losses from transport and handling, extra-EU imported waste, and the plastic waste generated from wastewater treatment plants, a total of 37,068 kt of plastic waste was generated in 2016 in the EU.

PCPW has different fates (Figure 1d). In 2016, it was estimated that only 20% of PCPW (7217 kt) was recycled, while the rest followed other paths; about 23% (8608 kt) was incinerated with energy recovery, and around 19% (6889 kt) was burned without energy recovery. Backfilling and reuse represent 137 kt and 183 kt, respectively. Landfilling is among the most adopted solutions for the disposal of 19% (6837 kt) of plastic waste. Around 9% (3400 kt) of PCPW entered the environment after being dispersed or mismanaged, accumulating and littering natural surroundings. The remaining PCPW (up to 10%, 3789 kt) was exported; some studies have pointed out that the lack of traceability of exported plastics may lead them to oceans [52]. Moreover, plastic waste sent to other countries is contaminated by other polymers of lower quality[51], hampering its reuse or recycling. Therefore, exported plastics may be summed, for the most, to environmental or landfilled PCPW.

According to Hsu et al. [51], the EU consumption of thermoset polymers in 2016 was equal to 11,140 kt. Moreover, 526 kt of thermosets was lost into the environment [53], potentially leading to 10,614 kt of collected waste. Once collected, this thermoset waste is typically unsorted [51] and, for the most part, is landfilled or used for energy recovery [54,55]. Mechanical reprocessing (through pulverization, hot compression, or the addition of adhesive) of thermosets into new products is possible but only before becoming a waste due to their contamination by other materials [55]. Therefore, ≈30% of PCPW is difficult to recycle with traditional methods due to its chemical structure and characteristics.

### 2.3. Options for Thermoplastic Waste Recycling

Different waste recycling strategies have been developed for thermoplastics. Specifically, following the circular hierarchy, waste recycling processes can be classified as primary, secondary, tertiary, or quaternary [46,56]. Primary recycling is used for the management of preconsumer plastic waste and consists of re-extrusion of the material. This waste is limited to fallout products, trimmings, or cuttings with a high level of homogeneity [56].

Secondary recycling (also named mechanical recycling or plastic reprocessing) is the most used method for thermoplastic waste recycling [47], although it often lowers the polymer quality [25,56]. The typical procedure for the secondary recycling process includes six main steps, which can be repeated and reordered: shredding of the collected plastic waste, washing, milling, separation and sorting (based on shape, density, size, color, and chemical composition), drying, and extrusion [47,57].

Tertiary recycling (also named chemical or feedstock recycling) consists of depolymerizing polymers into oligomers or monomers that are successively repolymerized or redirected to other applications such as fuels [46,58]. The main technological processes are gasification, pyrolysis, glycolysis, and hydrolysis [59]. These technologies are mainly used when the separated waste plastic fractions do not have sufficient quality to allow for fully mechanical recycling. For example, industrial waste (e.g., from the packaging industry) can be valorized with tertiary (or quaternary) recycling systems.

Quaternary recycling refers to incineration for recovery of the energy released during combustion (depending on calorific values of the material and on the plant configuration) to produce heat and electricity [59]. This process is considered the last option in the circular hierarchy, and it is typically used when previous recycling options are not applicable or for plastic materials that are difficult to sort.

Based on the type of the overall process, plastic recycling procedures can be further distinguished into closed- or open-loop processes [56]. In closed-loop recycling, the recycled materials are used to generate the same product from which they were originally recovered. The new product can be made up of a mixture of recycled plastic and virgin material. Closed-loop recycling is suitable for a small portion of plastic waste (e.g., PET packaging products) [47]. In open-loop recycling, the recycled materials are used for a product that is different from the original one. If the quality of the produced recycled material is higher than that of the original product, the process is defined as an upcycling process. In the other case, the process is defined as a downcycling process. Some examples include textile fibers made from PET bottles, printer components made up of polycarbonate from bottles or flooring tiles from mixed polyolefins [60,61].

### 2.4. Thermoplastics Recycling Issues: The Plastic Packaging Waste Case

The largest PCPW flow (≈50%) consists of plastic packaging waste (PPW). A recent detailed study on PPW in Europe [62] exemplified the amount and composition of thermoplastic leakages during the three stages between waste generation and recycling: collection, sorting at material recycling facilities (MRFs), and recycling in recycling plants (RECs) (Table 3).

Interestingly, the lowest value was the collection rate, with ≈60% of PPW never entering the recycling process, heading straight to other disposal methods (e.g., landfill, energy recovery, and incineration). Among the most diffused polymers, packaging made of PET was more collected than PP and PE. Once reaching MRFs, up to 30% (2002 kt) of collected PPW was rejected, while another 26% (1832 kt) was sorted for export (Figure 2). In this step, the highest leakages were for packaging made of PP, PS, and LDPE films (Figure 3a).

Exported plastics (Figure 3c) were mainly composed of PET and PE polymers and characterized by low polymer purity, which hinders their recyclability [51]. This implies that only <20% of total PPW effectively reached RECs, where another ≈22% (696 kt) was not recycled, mostly consisting of PS, PP, and LDPE films (Figure 3b). Globally, ≈15% of PPW was recycled, while the rest was rejected, exported, and otherwise disposed of, mainly during collection and sorting.

There are specific reasons for these losses of plastics before and after entering any recycling process. Currently, mainly high-value, monopolymer, rigid types of plastics are sorted for reprocessing [61], while plastics with low bulk density are often excluded. These plastics include LDPE films and bags. Another challenge of the sorting step is related to the limitations of near-infrared (NIR) sensors, the most used plastic sorting technology [64]. Such sensors are blind to black carbon materials because black pigments absorb the emitted light, preventing polymers from being correctly sorted. Black plastics were estimated to represent between 3–6% and 14–18% of total PPW volume and 18% of hard plastic [65]. Moreover, false sensor readings often occur, usually when only one layer of a multilayered object is scanned. As a consequence, contamination occurs between polymers that have different structures and processing melting points, leading to damage to recycling equipment and the deterioration of the properties of recycled materials in RECs [56]. This also occurs when NIR technologies are coupled with manual sorting (a common practice) or other plastic sorting systems that exploit the varying densities of polymers (e.g., sink-float and hydrocyclone processes); even if these systems ensure the achievement of high purity levels in downstream operations, especially for high-value plastics [47,56], they usually come at the cost of further rejects of low-quality collected plastic waste. This problem is not specific to PPW but affects any sector of PCPW, including textiles, which is a sector in which there has been a rapid growth of the production and environmental dispersions of materials made with fibers obtained by mixing different polymers. In particular, while mainly single-polymer products can be recycled at RECs, difficult-to-separate blended materials pose a demanding challenge [66].

### 2.5. Removal of Environmental Plastic Pollution

Plastics released into the environment are converted by abiotic forces into plastic debris of different sizes and can generally be divided into two categories: macroplastics, with a minimum size of 25 mm [67], and microplastics, which are smaller than 5 mm [68]. Microplastics are further divided into primary and secondary microplastics based on their original sizes [67]. Primary microplastics are designed and built to have microscopic size (e.g., plastics contained in cosmetic, cleanser, and scrubber formulations), whereas secondary microplastics are derived from macroplastic fragmentation both on land and in marine environments [69].

The amount of plastics dispersed into the environment in 2016 in Europe [51] was estimated to be ≈2264 kt of macroplastics (Figure 4a) and ≈1106 kt of microplastics (Figure 4b). The majority of scattered macroplastics is composed of thermoplastic polymers (PP, PE, and PAs), while car tires and road marking coatings are the major source of microplastics [70].

Some studies have evaluated whether recaptured dispersed macroplastics could be potentially recyclable; the recyclability of weathered PET bottles and HDPE caps was studied under simulated marine conditions [71]. After 3–4 years, HDPE caps were only slightly damaged (0.096% mass reduction) but still able to close properly, while PET bottles showed a decrease in transparency from 89 ± 3% to 43 ± 2%, making these products unsuitable for uses requiring transparency. Furthermore, tensile strength and Young’s modulus were not altered, whereas impact strength was reduced by 37%, and strain at break was found to be roughly halved. These quality parameters allowed mechanical recycling [71] after sorting and cleaning. This means that there are promising possibilities to recycle a portion of macroplastics collected from the environment, although the amount and quality of products is expected to be lower than from the recycling of collected PCPW.

For the removal of microplastics, the most mature technologies are based on the use of filters and have been applied in wastewater treatment plants (WWTP) [72,73]. A study conducted in Italy found that around 495 ± 61 microplastics per liter could be detected at the inlet of WWTPs, mainly between 100 and 499 μm in size. HDPE-LDPE (55%) represented the most common polymers, followed by PP (37%) and other less frequent polymers, such as PS (3%), PAs (2%), PET (1%), polyacrylonitrile (1%), and silicones (1%) [74]. The retention efficiency at WWTP outlets after filtration was higher than 94%. Disinfection (UV or chlorination) further reduced microplastic concentrations by up to 5.8 ± 2.7 microplastics per liter prior to discharge into water bodies [75]. Current European legislation does not indicate limits for microplastic contents. Little information is available on the effects of discharged microplastics on human health.

Apart from this common treatment process for water bodies, knowledge on microplastic remediation by means of physical, chemical, and biological (microbes) approaches was recently reviewed [76,77,78]. Research is mainly focused on testing the use of chemicals for microplastic depolymerization, the creation of composting sites under controlled conditions, and on the analysis of microbial diversity and chemical composition changes in natural sediments contaminated by microplastics. Major concerns associated with the application of in situ remediation strategies are related to the mixed and heterogeneous composition of microplastics, the use of chemicals and radiation in natural ecosystems, and the largely incomplete (<15%) and relatively slow biodegradation reported in composting sites [79,80,81]. Collectively, the proposed methods for microplastic remediation are still far from any large-scale application.

## 3. Biotechnological Systems Applied to Plastic Depolymerization to Target Recycling

Biotechnology tools have been employed for more than 20 years to depolymerize complex biomass waste into organic building blocks that, in turn, can be converted into useful intermediate feedstocks. These processes are typical applications of the circular (bio)economy concept. One of the most interesting examples is the conversion of lignocellulosic materials contained in plant waste from farming activities into bioethanol [82,83]. More recently, biotechnology strategies based on the use of microbial enzymes have been applied also to the degradation of petroleum-based plastics with the aim of providing greener alternatives to the traditional methods of recycling (Table 1).

Various enzymes are used by bacterial or fungal organisms to fragment polymers into units of about 10–50 carbon atoms, which are used as carbon sources [84]. The catalytic capacity of highly purified stocks of enzymes (produced at an industrial level) can be exploited in biochemical processes performed under mild conditions (e.g., water solution in the 20–80 °C range) because they are biocatalysts evolved to act under conditions compatible with life. This allows for lowering of the economic and environmental costs of traditional processes, which usually require high temperatures and pressures [27]. The wide diversity in the metabolism of living organisms allowed for the discovery of biocatalysts suitable for many chemical reactions, including those responsible for plastic degradation, although consortia of whole organisms may also be required in some cases [85]. The enzymatic or microbial recycling of plastic materials is divided into three main steps: (i) the production of the biocatalysts, (ii) the depolymerization reaction of the polymer into monomers, and (iii) the purification and reuse of the monomers for resynthesis of the original material (closed-loop processes) or their bioconversion into different products (open-loop processes) [36].

A plethora of options is available for open-loop routes leveraging biotechnologies, as enzymes can catalyze multistep reactions to transform monomers into completely different molecules, which possibly have a value (per mass unit) superior to that of the original polymeric material [86]. The processes for open-loop plastic recycling can be performed in different bioreactors (employing purified enzymatic cocktails or whole organisms) or in a one-pot system [36]. The main classes of enzymes involved in plastic degradation and their products are reported in Figure 5. Table 4 reports the best-performing bio-based plastic depolymerization systems, which are predominantly implemented at laboratory scale.

### 3.1. Enzymatic PET Recycling: Potential Industrial-Scale Application

#### 3.1.1. Enzymatic Degradation of PET

PET is a semicrystalline material made up of polymeric chains of alternating terephthalic acid (TPA) and ethylene glycol (EG) joined by ester bonds. In crystalline regions, the polymer chains are tightly packed in parallel, while in amorphous regions, the chains assume a disordered conformation. The degree of crystallinity depends on the production process and affects the chemophysical properties of the polymer which, in turn, are related to the final product performance. For instance, PET used in the production of bottles and textile fibers has a high crystallinity (between 30–40%), while PET is mainly amorphous in food packaging (crystallinity < 10%).

Research on biotechnological degradation of PET started in 2005 [117]. Since then, several microbial PET-hydrolyzing enzymes (PHEs) have been discovered for their ability to break down PET into mono-(2-hydroxyethyl)-terephthalate (MHET), bis-(2-hydroxyethyl)-terephthalate (BHET), TPA, and EG by hydrolyzing the ester bond [118] (Figure 5). Examples of the most active enzymes are reported in Table 4. These enzymes descend from the same ancestral carboxyl ester hydrolase of the GxSxG family [119]. They share the same catalytic mechanism and span different enzymatic classes specialized on different natural substrates: cutinase (EC 3.1.1.74), lipase (EC 3.1.1.3), and PETase (or PET hydrolase) (EC 3.1.1.101) [120,121,122]. Remarkably, the PETase class was created in 2016 after the discovery of a PHE from *Ideonella sakaiensis* (IsPETase), a bacterium isolated near a WWTP, with optimal PET degradation activity at 40 °C [123]. IsPETase and other *I. sakaiensis* enzymes were identified and shown to allow for the depolymerization and metabolic assimilation of PET degradation products as a carbon source [124]. The relevance of such a PET waste assimilation process in natural environments was questioned because the depolymerization of highly crystalline PET (such as that used to produce bottles) under natural conditions for the growth of the microorganism (30 to 40 °C) is highly inefficient [125,126]. Indeed, highly crystalline PET regions and low temperatures (<40 °C) greatly reduce the accessibility of PET chains to PHEs. Accordingly, the accessibility is maximal between 70 and 74 °C, near the glass transition temperature of bulk PET [127,128].

Consequently, known PHEs seem unsuitable for efficient degradation of untreated highly crystalline PET at room temperature [128,129]. This prompted a search for thermostable PHEs produced by protein engineering [87,90,91,93,130,131] or enzyme discovery [92,132,133,134]. Among such attempts, the company Carbios, in a public–private partnership with the Toulouse Biotechnology Institute and other global partners, was the first to produce a thermostabilized engineered variant of the leaf–branch compost cutinase (LC-cutinase) suitable for an industrial-scale recycling process of PET in 2020 [135]. Carbios built a demonstration plant in September 2021 with a business plan to extend the system to other companies under a paid license agreement by 2023 and revenues expected by 2025 [136].

The Carbios technology and other promising enzymatic PET recycling processes (Table 4) act on low-crystallinity submillimeter PET films [91,92,93,137] or amorphized/micronized post-consumer PET materials (C-ZYME^®^, based on the enzyme variants reported by Tournier et al. [90] and various patents (i.e., WO2017198786A1, EP 3517608A1, and WO2020021118A1)). In practice, post-consumer PET degradation processes were operated after sorting, washing, grinding, and amorphization. These pretreatment steps were already established for non-enzymatic recycling processes [138]. The step involving the use of enzymes was designed to be performed in <1 day in a buffered water suspension (pH between 7 and 9) inside a bioreactor containing a preprocessed polymer and close to the PET glass transition temperature. However, enzymatic incubation above 70 °C does not permit complete degradation of the amorphized PET due to an ageing process that results into polymer recrystallization after <10 h [139,140]. Recent studies [140,141,142] revealed relatively high degradation rates, even when performing the reaction at lower temperatures (50 to 60 °C), despite requiring more time (≥1 day). These milder conditions were also tested on untreated and non-micronized low-crystallinity post-consumer PET materials [92,93]. Moreover, Tarazona et al. recently demonstrated that a reduced PET thickness and an increased accessible surface area reduce the glass transition temperature by up to ≈40 °C (for nanometric film) [89]; while this can speed up PET degradation at lower temperatures and inhibit recrystallization, it does not block degradation-induced thermal inactivation, which consists of the accumulation of inactivated protein aggregates on the PET surface, causing the inhibition of further polymer degradation (a process called *passivation*).

Overall, promising industrial-scale PET degradation technologies require a buffered water suspension at ≥50 °C and PHEs engineered to be stable, efficient, and with low product inhibition (eventually by combining multiple enzymes) allowing for complete depolymerization as fast as possible in order to prevent PET recrystallization and *passivation* [89]. A recent work showed significant condition-dependent differences in the depolymerization process (buffer, pH, enzyme, and substrate loading) catalyzed by similar engineered PHE variants [141]. In addition, the enzymatic capacity to adsorb and turn-over during PET chain hydrolysis were recently observed to follow a tradeoff [143]. For these reasons, the biochemical kinetics and the mechanism of enzymatic PET degradation should be studied in detail to select the best enzymes and reaction conditions under which both the PHE activity is optimal and the PET polymer is most degradable, prior to scaling up the process [144]. Moreover, it is critical to set up a standard assay condition to compare different PHEs (different conditions are used for the enzymes reported in Table 4). As an example, a recently thermostabilized IsPETase (HotPETase) was shown to have initial degradation rates at 60 °C, much higher than the ICCG LC-cutinase used by Carbios on low- and high-crystalline materials (including a composite PET-PE film); however, it mainly produced MHET (TPA is the main product of ICCG LC-cutinase) and was inactivated after few hours on highly crystalline substrates, before the degradation was completed [94]. Furthermore, L92F/Q94Y PHL7 cutinase was more efficient than ICCG LC-cutinase on low-crystalline materials, while it was less active on highly crystalline PET powder [91]. Finally, different relative degradation rates on PET of different sizes and crystallinity were also measured when comparing many recently discovered PHEs [145], motivating the possible need for a cocktail of multiple PHEs for the degradation of PET materials of different crystallinity and subjected to different pretreatments.

#### 3.1.2. Closed-Loop PET Recycling by Enzymatic Depolymerization

Carbios company demonstrated the feasibility of a closed-loop recycling process by showing that it is possible to use the monomers (TPA and EG) produced from PET depolymerization of post-consumer PET flakes to produce a PET bottle possessing the same properties as a bottle produced from virgin PET [90]. This process may also be fostered by monomers recovered from PET contained in textile polyester waste; an enzymatic cocktail of PHEs and other enzymes could be added to chemical treatments (in concert or in a sequential manner) to allow for separation and to reach a superior depolymerization yield of polymers without causing the high material losses associated with traditional methods. This is because the PET contained in fibers of commercialized textile products is often blended with other natural (i.e., cellulose and cotton) or synthetic (polyesters, nylons, elastane, etc.) polymers [66,146].

A recent work used Carbios data to conduct a technoeconomic analysis (TEA) of a closed-loop PET recycling process from bottle-derived post-consumer flakes [138]; a minimum production cost (MPC) for TPA of USD 1.93 kg^−1^ was estimated, most of which (>50%) was due to the price of petroleum-based PET flakes (feedstock PET) acquired from MRFs, the plastic pretreatment, and the recovery of the degradation products and co-products. In contrast, the step involving enzymes accounted only for 20% of the cost. Nevertheless, the model shows that besides the feedstock PET price, which is an external factor, the load of PET in the degradation solution and the rate of enzymatic conversion are the other dominant factors affecting MPC, enabling a reduction in cost of up to USD 1.60 kg^−1^ [138], which is less than double the average petroleum-based PET price in 2021 (up to USD 0.85 kg^−1^) [147]. It must be pointed out that this TEA estimated that the environmental impact of TPA obtained from the enzymatically based recycling process is nearly halved with respect to virgin TPA production. A recently published life cycle assessment (LCA) based on the USA economy confirmed the previous estimation of the production costs [138] but indicated that current enzymatic PET recycling technology exhibits a 1.2 to 17 times superior environmental impact relative to virgin polymer production [148]. NaOH usage, PET waste collection/shredding/amorphization, and electricity were estimated to be the highest-impact contributors. Sensitivity analysis indicated that, through interventions during all stages of the recycling process, from feedstock preparation to product recovery, it is possible to achieve an environmental impact similar or even lower than virgin polymer production. However, a superior enzymatic depolymerization efficiency without changing any other condition was estimated to contribute only up to 14%. Main contributions were estimated to be provided by minimization of PET losses during sorting, elimination of PET flake pretreatments, enzyme immobilization, and reaction buffer reuse. A major environmental benefit should originate from the reduction in NaOH consumption by using completely different buffers and PET loadings to facilitate TPA precipitation or by using more environment friendly (e.g., ammonia) or recoverable bases. Overall, this study suggested that further process optimization is required to minimize waste during feedstock preparation and product recovery.

Novel biocatalysts should allow for effective PET depolymerization under different buffer conditions (including organic solvents) of less pretreated PET-containing materials rather than showing a superior degradation rate under a broad range of pH and temperature conditions. Moreover, this process should be evaluated on different sources of PET waste, including those from commonly mixed products (i.e., textiles) for which traditional technologies are less effective and for which both the high crystallinity and the material substructure are expected to pose accessibility challenges to degradative enzymes [66]. The goal is the setup of a closed-loop enzyme-based process with a halved environmental impact compared to virgin polymers, compensating for the superior costs of PET produced from the enzymatic recycling process with an increased environmental gain.

#### 3.1.3. Open-Loop PET Bio-Recycling

The use of biotechnological tools into open-loop processes could represent an opportunity to overcome the overpricing problem of closed-loop enzymatic PET recycling; enzymes and/or microorganisms can be used to assimilate and metabolically convert PET building blocks (TPA and EG) into chemically different intermediate bulk compounds or polymers with high added value [58]. This could also ensure the economic sustainability of treating low-quality plastics collected from contaminated environments. Several processes have been tested at the laboratory scale (Table 5).

Kim et al. [149] presented a two-organism system for the biological conversion of chemically depolymerized PET; TPA was converted into protocatechuic acid (PCA) and other PCA-derived compounds (i.e., gallic acid, pyrogallol, catechol, muconic acid, and vanillic acid) by an engineered *E. coli* strain, while *G. oxydans* KCCM 40109 was used to produce glycolic acid from EG. These compounds are important bulk chemicals to produce inks, bioplastics, flavorings, and fragrances. The same authors also demonstrated the possibility of using TPA produced by the enzymatic depolymerization of PET to produce catechol for the production of multifunctional coating materials [150]. Kang et al. [151] proposed the use of an engineered *E. coli* strain to convert TPA monomers from post-consumer PET into 2-pyrone-4,6-dicarboxylic acid, a valuable monomer for the synthesis of next-generation biodegradable plastics. A *Pseudomonas putida* KT2440 strain was engineered to use BHET (a different PET biodegradation product) to produce β-ketoadipic acid, a building block used to produce performance-advantaged nylons [152]. A one-pot system was proposed by Tiso et al. for the sequential enzymatic depolymerization of PET into EG and TPA [153], followed by *Pseudomonas umsongensis* GO16 conversion into intracellular polyhydroxyalkanoates (PHAs) and extracellular hydroxyalkanoyloxy alkanoate, which is used in a chemical copolymerization reaction to produce a novel bio-based poly(amide urethane) PUR. A similar system based on the use of an *E. coli* strain was proposed by Sadler et al. to produce vanillin flavor [154]. Liu et al. [155] proposed the first one-pot microbial recycling system from PET using two co-cultivated engineered organisms: a *Yarrowia lipolytica* Po1f strain expressing a PHE to depolymerize PET at low temperature and a *Pseudomonas stutzeri* strain to convert TPA into the bioplastic polyhydroxybutyrate. The same group showed that muconic acid can be produced with a single engineered *Pseudomonas putida* strain [156].

**Table 5 ijms-24-03877-t005:** List of recent open-loop microbial conversion attempts starting from PET and polyester-PUR degradation products. The different approaches are distinguished with respect to the processing strategies proposed in [36]. Abbreviations: EG—ethylene glycol, AA—adipic acid, BDO—1,4-butanediol, TPA—terephthalic acid, BHET—bis(2-hydroxyethyl) terephthalate, PHAs—polyhydroxyalkanoates, HAA—hydroxyalkanoyloxy alkanoate, PHBS—polyhydroxybutyrate.

Approach	Enzymes or Microbial Strain(s)	Plastic Monomer	Conversion Products (Yields %)	Reference
Separate depolymerization and conversion	*Pseudomonas putida* strains JM37 and KT2440	EG	Glyoxylic acid (≈20%)	[157]
	Three derived *Pseudomonas putida* KT2440 strains	AA, BDO, EG	Monorhamnolipids (<1%)	[158]
	*Escherichia coli* strain	TPA, EG	Gallic acid (≈93%), pyrogallol (≈40%), muconic acid (≈85%), vanillic acid (≈40%), glycolic acid (≈98.6%)	[149]
	*Escherichia coli* strain	TPA	2-pyrone-4,6-dicarboxylic acid (96.08%)	[151]
	*Pseudomonas putida* KT2440 strain	BHET	β-ketoadipic acid (≈76%)	[152]
	*Escherichia coli* strain	TPA	Catechol (≈67%)	[150]
Combined depolymerization and conversion	LCC ^1^, *Pseudomonas umsongensis* GO16 strain	TPA, EG	PHAs and HAA (<2%)	[153]
	LCC ^2^, *Escherichia coli* MG1655 RARE strain	TPA, EG	Vanillin (≈79%)	[154]
Consolidated bioprocessing	*Yarrowia lipolytica Po1f* expressing PETase ^3^, *Pseudomonas stutzeri TPA3*	TPA, EG	PHB (≈2.5%)	[155]
	*Pseudomonas putida* KT2440-tacRDL expressing LCC ^4^	TPA, EG	Muconic acid (≈50%)	[156]

^1^ Wild-type enzyme from [135]. ^2^ WCCG variant from [90]. ^3^ Variant from [159]. ^4^ ICCG variant from [90].

These examples of open-loop PET recycling employing enzymes and microorganisms represent proof-of-principle experiments demonstrating the potential of microbial enzyme-driven upcycling processes.

### 3.2. PUR Biodegradation Is Restricted to Polyester-Based Polymers

#### 3.2.1. Enzymatic Degradation of PURs

PURs represent the largest fraction of plastic waste that is unrecyclable using traditional technologies. PUR polymers are formed by many different subtypes of two building blocks belonging to polyol (e.g., 1,4-butanediol, BDO) and isocyanate (e.g., toluene diamine, TDA) classes linked by urethane bonds (Figure 5). Moreover, isocyanates usually contain aromatic rings that increase the rigidity of the polymer and are covalently crosslinked to polyols of other chains by the formation of ester bonds in polyester-type PURs or by the formation of ether bonds in polyether-type PURs [160]. PURs are among the most diversified polymers, with different compositions in bonds, building blocks, and crosslinking distributions. This permits the generation of PUR materials with tailored physical properties [161]; however, the consequence is that a material-specific toolbox of enzymes with different activities is required for their degradation [162].

The enzymes identified to date to cause some degradation or weight loss of PURs are mainly promiscuous esterase (EC 3.1), urease (EC 3.5.1.5), protease (EC 3.4.21), and amidase (EC 3.5.1.4) enzymes [161]. Some esterases were shown to potentially degrade liquid dispersion (Impranil-DLN^®^) or bulk polyester-type PURs, but there is no evidence of their capacity to cleave urethane bonds [161]. Proteases and amidases have been shown to hydrolyze both urethane and ester bonds, but very low degradation rates were reported on commercial elastomer polyether-type PUR films or polyester-type PUR pellets [163]. A combined system involving an esterase and an amidase showed improved synergistic degradation (33%) after 51 days on a model PCL-based thermoplastic PUR [100]. A laccase-mediated (EC 1.10.3.2) system previously reported to degrade PE caused a reduction of a few percentage points in weight of lab synthesized polyether-type PURs in 18 days [164]. Urethanases (EC 3.5.1.75), which are enzymes capable of specifically hydrolyzing the urethane bond between polyols and isocyanate units, were recently discovered [165]; although their use for full depolymerization of PURs within a few days is very promising, these enzymes were used and demonstrated to be effective only on the urethane bonds in low-molecular-weight dicarbamate aromatic diamines obtained after the chemical depolymerization (by glycolysis) of non-urethane bonds in a polyether PUR foam. Therefore, this process is bio-based only in the last step.

Overall, even if enzymes capable of degrading the urethane and ester bonds in PURs have been discovered, processes that make exclusive use of enzymes have relatively high degradation rates only on synthetic or oligomeric PURs designed to be more biodegradable [166,167]. Therefore, the best options for PUR biodegradation are the use of environmental microbial strains or consortia, the design of more degradable novel PUR materials, or synergy with chemical methods of depolymerization [168].

#### 3.2.2. Microbial Degradation of PURs

Some promising microbial degradation systems acting on polyester-type PURs have been tested at lab scale in the last 10 years. Impranil-DLN^®^ was used as the sole carbon source by *Pseudomonas putida* A12, achieving 92% degradation within four days under mild conditions [97]. *Bacillus subtilis* MZA-75 and *Pseudomonas aeruginosa* MZA-85 were able to co-act on a polyester-type PUR film, resulting in 40% weight loss after 30 days [103]. Examples of polyester-type PUR biodegradation were also reported for fungal strains, both on Impranil-DLN^®^ [98], a liquid varnish [101], and a commercial film [102]. Oligomeric fragments are usually reported as degradation products, which means that complete hydrolysis to single building blocks was not achieved. Moreover, microorganisms usually require other carbon sources to grow on PURs and may be susceptible to the accumulation of metabolically toxic plastic degradation products or additives, as observed for a *Pseudomonas* strain that released diamines from degraded PURs [158].

Polyether-type PURs show higher recalcitrance to microbial biodegradation, in particular when rigid aromatic polyisocyanate units are present, since they are less accessible and not susceptible to degradation by esterase activity [160]. Until now, only three studies have reported a significant biodegradation of polyether-type PUR materials: *Tenebrio molitor* larvae were shown to ingest polyether PUR foams efficiently, resulting in a significant mass loss of nearly 67% after 35 days. However, polyether PUR fragments were found in the frass of the larvae, indicating partial degradation [99]. *Alternaria* sp. strain PURDK2 caused 27.5% degradation of polyether-type PUR cubes after 10 weeks [104], and some *Cladosporium* fungal strains were reported to cause about 65% weight loss of a polyether-type PUR foam in 21 days [101].

Considering the reported degradation rates, the use of simple ad hoc synthesized or non-waste PUR products, and the lack of knowledge on the complex enzymatic machinery produced by microbial consortia [100], it is not currently possible to depict a practical industrial-scale biodegradation process, even for less recalcitrant polyester-type PUR waste. Moreover, solid polyester-type PURs and polyether-type PURs were only partially biodegraded (up to 50% at most), requiring a few weeks or even a few months. Therefore, for more recalcitrant PURs, a manly bio-based depolymerization process is still unfeasible.

In the next few years, it will be crucial to isolate the microbial enzymatic machinery involved in the degradation of less recalcitrant polyester PURs by using model PUR substrates and developing new specific assays (as presented by Liu et al. [169]). This will allow for mimicking of the whole-organism biodegradation action using an enzymatic cocktail or a stepwise addition of enzymes. Future investigations should also be focused on specific commercial PUR materials [161].

#### 3.2.3. Closed-Loop and Open-Loop PUR Bio-Recycling

The complex structure of PURs leads to a variety of degradation products, including organic acids, organic alcohols (e.g., EG), and diamines (Figure 5). Carboxylic acids and alcohols can be used for the synthesis of virgin PURs and other poly(ester–ether–urethane) materials [168]. Amines such as toluene diamine (TDA) can be recovered and used to directly synthesize polyamides or toluene isocyanate (TDI) to subsequently make virgin PURs (bio-based closed-loop recycling). Although the full depolymerization of a polyether-PUR foam was demonstrated with a chemoenzymatic approach [165], its application on a real waste PUR material and the use of the degradation products (polyether-polyols, diethylene glycol, and aromatic diamines) for the resynthesis of a waste-like PUR material was never attempted.

Some PUR degradation products have also been tested as microbial carbon sources (Table 5). EG can be used as a substrate to produce PHAs, which can be used to manufacture biodegradable medical devices, as well as glyoxylic acid, which is usually used for buffering solutions and other products to treat human skin and hair [157]. Recently, polyols (adipic acid (AA), 1,4-butanediol (BDO), and EG) were used as a carbon source for a mixed culture of three recombinant *Pseudomonas putida* KT2440 bacterial strains [158]. These bacteria can produce rhamnolipids, which are specialty biosurfactants applied in many industrial sectors. Coupling this process with an efficient extraction of toxic aromatic diamines contained in PURs (i.e., TDA), as proposed by Chen et al. [170], can be used to realize a green route for the open-loop bio-recycling of PUR degradation products.

### 3.3. Prospective for the Biological Degradation of Other Petroleum-Based Plastics

The biodegradation of other recalcitrant petroleum-based plastics (PAs and plastics with only C–C covalent bonds between their building blocks, i.e., PE, PS, PP, and PVC) is challenging; no degradative enzyme acting on the hydrolytic cleavage of the C-C bond is known. Moreover, some of these plastics (PE, PP) are completely crystalline, preventing enzymes from accessing the potential attack sites. Therefore, a multistep reaction catalyzed by a combination of abiotic pretreatments and different types of oxidative enzymes has been proposed [171].

There are many different types of PAs (polymers made of repeating units of aliphatic, semi-aromatic, or aromatic molecules linked via amide bonds), the most popular being nylon and Kevlar [172]. Since natural silk is also a PA from a chemical point of view, it was expected that enzymes able to degrade synthetic PAs do exist in nature. Unfortunately, to date, no known microorganism able to effectively degrade polymeric PAs has been identified, while it was demonstrated that several bacteria can act on short oligomers of linear or cyclic nylon and are possibly useful for recycling the byproducts of PA production [85,173]. A more recent study by Biundo et al. [174] demonstrated that it is also possible to engineer PHEs to abolish PET degrading activity and simultaneously increase the catalytic efficiency towards soluble PA oligomers. A manganese-dependent peroxidase from white rot fungus IZU-154 is the only enzyme reported to act on high-molecular-weight nylon fibers [175].

It was proposed that PE biodegradation only happens after abiotic (ultraviolet exposure) or enzymatic oxidation by different classes of oxidoreductase (alkane hydroxylase EC 1.14.15.3, laccase or multi-copper oxidase EC 1.10.3.2, peroxygenases EC 1.11.2, or Mn-peroxidase EC 1.11.1.13). This results in the introduction of aldehydic, ketonic, or alcoholic groups, which allows for subsequent processing by other enzymes. The latter can produce intermediate acid oligomers that are intracellularly taken-up by microorganisms and further metabolized [176,177]. Several microbial strains capable of degrading PE with this proposed system have been reported, including the gut microbiota of invertebrates [106,171]; animal phenol oxidase (EC 1.14.15.3) enzymes contained in the saliva of wax worm (*Galleria mellonella*) larvae were also demonstrated to be involved in the oxidative degradation of PE [107]. No system studied to date can achieve complete biodegradation without abiotic intervention. Photo-oxidative and thermal pretreatments, although expensive and technically demanding, were shown to oxidize and release oligomers from PE, improving their biodegradation rates [178]. In some cases, PE degradation of up to 30% was reported in one month [109,111], with even nearly complete bio-assimilation over >8 months [179]. In the absence of such pretreatments, the highest biodegradation rates, determined as polymer weight loss, were observed for LDPE films or bags (up to 10% within a few days to one month) [105,108,171]. These degradation rates were reported only for LDPE materials, while HDPE is less accessible to microorganism attack and much more recalcitrant [180].

Although PS is amorphous, its C–C backbone is more resistant to enzymatic cleavage than PE [36]. No enzyme has been identified with the ability to efficiently degrade high-molecular-weight PS polymers, despite some studies linking PS degradation to the same laccase and oxidoreductase enzymes associated with PE degradation [181]. Some organisms [113,114], including invertebrates assisted by their gut microbiome [182], were reported to cause significant weight loss in pretreated PS material, with rates of around 10% over one month. Another study reported that *Pseudomonas putida* CA-3 cultures can reach this degradation level in 8 days and produce PHAs. In this case PS was thermally pretreated at an elevated temperature (pyrolysis) to convert it to a styrene oil; this two-step process is very energetically demanding [112].

PP polymers can be present in three stereoisomeric forms, namely atactic, isotactic, and syndiotactic [183]. All PP forms are more resistant than PE to heat and to chemical attack, including the action of enzymes [178]. Some studies of PP biodegradation reported weight loss rates of about 10% per month. In all cases, these processes required a strong pretreatment with either γ- or UV-irradiation or heat or the use of polymer blends mixing PP and carbohydrates [115,116,184].

Finally, PVC, a synthetic polymer that contains chloride, is a form of waste that poses a serious pollution problem [185]. Very few studies have reported the biodegradation of low-molecular-weight PVC oligomers or film [178,186]. The capacity to degrade PVC was associated with the activity of catalase-peroxidase (EC 1.11.1.21) enzymes in white rot fungi [187] or from bacteria found in the gut of insects larvae [186]. Nevertheless, the use of microorganisms to remove PVC waste is still controversial because of the possible release of organochlorine compounds that may be highly toxic for the organisms themselves and the environment [178].

The fatty acids released by the oxidative bio-depolymerization of C-C plastics cannot be used in a closed-loop process of recycling, but they could be used as a carbon source for microbial growth to produce high-added-value products. While an open-loop fully bio-based process has not been established, in a recent proof-of-principle study, an open-loop recycling process was designed to produce β-ketoadipate or PHAs using microbial strains growing on organic acids obtained from chemically oxidative depolymerization of mixed resins or post-consumer waste (made with PS, HDPE, and PET) [188]; although promising, an LCA analysis is required to assess the industrial-scale feasibility of the proposed systems. A recent LCA evaluation focused on the use of fatty acids from chemically depolymerized and oxidized PE as a carbon source for growing a *Pseudomonas putida* KT2440 strain to target PHA production: the proposed approaches were more expensive or had a greater impact on the environment than waste-to-energy approaches [189].

## 4. Perspectives

### 4.1. Plastic Recycling Challenges in the EU: Current Limits of the Biotechnological Approaches

European targets for plastic waste recycling rates are set by Directive 94/62/EC to 50% and 55% by 2025 and 2030, respectively. These targets are stricter for PPW, requiring that 65% and 70% of PPW be recycled by 2025 and 2030, respectively [190]. These targets can be achieved by summing the already recycled PCPW (≈20%) with the fraction (18,514 kt y^−1^, ≈50% PCPW) that we estimated to be potentially included in future recycling flows (Figure 6a). The polymer composition of this unrecycled waste plastic flow is reported in Figure 6b.

The main contribution (57%) of this fraction is thermoset plastics (29.4% PURs and 27.9% other thermosets) (Figure 6a). Therefore, the most urgent interventions should focus on finding alternative recycling options to allow for the production of virgin thermoset materials from waste.

Thermoplastic rejects from MRFs or RECs account for 25% of this unrecycled PCPW flow (Figure 6a). Exported plastic waste (10%) was included, as the European Commission has stated that all PPW should be recycled in the EU market by 2030 after the Chinese government’s ban (April 2017) on imported recyclable solid waste [191]. An improvement in the amount of recycled plastic should be derived from improved sorting and preprocessing to reach higher purity of PCPW fractions at RECs; this is achievable by coupling traditional sorting technologies (i.e., manual sorting and NIR sensors) with machine-learning-based recognition of plastic waste from images [192,193]. This is expected to only increase the recycling rate in RECs, while thermoplastics rejects generated by mixed and small-particle plastics would remain an issue. This suggests that a fundamental change in the plastic value chain is required for marketable plastic materials to account for end-of-life circularity options in waste treatment processes, especially during the sorting step.

Although environmentally dispersed plastics are reported to be the smallest flow of unrecycled PCPW (18%, Figure 6a), they also represent the most complicated fraction to be recaptured (microplastics) and valorized (the quality of macroplastics recovered from the environment is poor). This flow should be lowered as much as possible by improving plastic collection rates through a focus on public behavior and municipal policies aimed at decreasing environmental leakages and progressively increasing the amount of collected plastics to target recycling.

The remining PCPW mixed with non-plastic materials and disposed of as non-plastic unrecyclable waste (targeted to landfilling, incineration, or recovery) was not considered as a potentially recyclable flow.

Considering the necessary interventions, microbial enzyme-based biotechnology could be integrated in the waste management sector and contribute to the introduction of new depolymerization methods for thermosets and rejected thermoplastics, allowing for resynthesis of virgin polymers or the open-loop upcycling of degradation products. Therefore, it is necessary to make biotechnological solutions available that are suitable to treat plastic waste of different quality, including materials made of (or blended with) many different polymers (not necessarily synthetic) that are difficult to separate with traditional methods prior to recycling; moreover, such systems should retain market competitiveness and an environmental impact equal to or lower than that of existing technologies. In this regard, the actual contribution of biotechnology to plastic degradation is limited, as only the bio-based depolymerization systems described for amorphous aliphatic polyesters (such as pretreated PET) and liquid polyester-type PURs have the potential to be exploited at the industrial scale, although not competitive with traditional approaches at present. Moreover, PET is, by far, the most consumed polyester; however, PET waste represents only a minor fraction (≈7%) of unrecycled flows with respect to PURs, thermosets, and other non-polyester plastics (Figure 6b).

A synergy with promising methods of abiotic secondary and tertiary PUR depolymerization approaches could allow for depolymerization of recalcitrant PUR waste, but no method has been tested on real PUR waste to date [49,165,168,194,195].

PAs and C-C backbone plastics, such as PE and PS, are currently much less efficiently biodegraded. Moreover, some of these plastics (i.e., PVC) may release toxic components after their degradation. Although biodegradation could be enhanced by new methods of abiotic preprocessing in the presence of organometallic catalysts, these treatments require a lot of energy and relatively high temperatures (>150 °C), thus contributing to higher costs and environmental impact [36,196]. Moreover, as most biodegradation processes have only been conducted at laboratory scale [176,178], without biochemical information on the kinetics of the involved reactions, with scarce data on degradation products, and often observing that mainly additives were cleaved [197], a fully bio-based depolymerization of non-polyester petroleum-based plastics is predicted not to be economically and environmentally competitive with traditional approaches within a 10-year time frame. Instead, it is necessary to look at new approaches of tertiary recycling [34,198,199] and to use them in combination with microbial enzymes to convert degradation products into high-added-value chemicals [188,200,201].

### 4.2. Finding New Biotechnologically Degradable Plastic Materials

Biodegradable bio-based plastics made (at least partly) with polymers and monomers that are biomass-derived are considered a future investment to reach plastic circularity [202]. Since first-generation bio-based plastics can be biodegraded only over long time periods [203], highly biodegradable polymers have been developed, such as protein-based biofilms from plant crops seed extracts [204,205] and from milk [206]. Moreover, attempts have been made to produce biofilms from the proteins of insects reared on organic waste to overcome the competition with the market of plant- and animal-derived food products [207]. It is important to report that lab-scale experiments are ongoing to synthesize more biodegradable thermoset materials, starting from mixing agriculturally derived oils with traditional precursors of polyester-type PURs [166,208,209,210,211]. The advantage of these biodegradable materials is that they can be assimilated as a carbon source by environmental organisms or at dedicated composting sites without releasing microparticles and without the need to develop a specific biotechnology in an industrial setup. On the other hand, there is an obvious tradeoff between biodegradability and durability; highly degradable bio-based plastics are highly sensitive to water and biotic factors, making them inadequate for high-end applications and as large-scale commercialized materials [212]. Therefore, new solutions to increase the lifespan of biodegradable biopolymers must be found.

It is known that several enzymes, including PHEs, can degrade the ester bonds present in petroleum-based (PET, polyester-type PURs, polybutylene adipate terephthalate (PBAT), and polycaprolactone) and bio-based (PHAs, polylactic acid (PLA), polyethylene furanoate (PEF), polybutylene succinate (PBS), or starch–cellulose blends) aliphatic polyester thermoplastics [213,214,215,216,217]. Although some of these polymers were found to be biodegradable (e.g., PLA and PBAT) and have already been used to produce packaging material and composting bags at industrial scale [218], they have been shown to require months to degrade in the natural environment and represent only about 0.4% of total plastic production. In addition, they are often used in blends with traditional non-biodegradable plastics to produce composite materials, resulting in the release of microparticles of the recalcitrant polymer [34,219]. Therefore, it is crucial to develop new plastic products with particular attention on aliphatic polyesters (i.e., PEF, PBAT, and PBS) to replace traditional recalcitrant petroleum-based plastic waste. Regardless of whether these plastics are biodegradable or bio-based, we argue that the most important aspect is the creation of suitable technologies to allow for their bio-based degradation and circular production from waste within an industrial setup, while avoiding environmental leakages.

### 4.3. Study of New Polyester-Degrading Organisms and Enzymes

Aliphatic polyesters represent valuable plastic polymers that could be, in principle, depolymerized by microbial enzymes; however, apart from PET, limited studies are available on the specificity and efficiency of polyester-degrading enzymes and organisms in nature [217]. Recent studies investigated whether plastics spread in the environment stimulated the evolution of polyester degradation and assimilation of molecular machineries in microorganisms [220,221]. The collected evidence is not yet conclusive; the observed alteration of microbial diversity and genetic content could be an adaptation in response to the effects of the presence of toxic compounds rather than an adaptation to the use of plastics as a carbon source [222]. As an example, a PHE produced by a bacterium from human saliva was recently discovered and shown to be much more active than IsPETase [134], the latter being a PHE isolated from an organism inhabiting highly plastic-polluted areas near a WWTP [124]. Moreover, Erickson et al. recently showed that PHEs are spread across many different taxa and that some enzymes have different catalytic residues and/or the presence of accessory polymer binding domains compared to traditional PHEs [145]. This suggests that PHE enzymes are likely promiscuous and that their capacity to efficiently depolymerize anthropogenic PET polymers under certain conditions is a catalytic side activity [33,223].

As a consequence, biodiversity could represent a valuable source of pluripotent biocatalysts with different specificities and efficiencies on different types of polyesters [85,145], motivating the search for plastic-degrading enzymes in nature. Possible approaches for this purpose were recently reviewed by Zhu et al. [41]. Traditional methods rely on collecting plastic samples from polluted areas or MRFs/RECs to isolate pure microbial cultures able to degrade model polyesters. Once a polyester degrader microorganism is found, DNA sequencing of these microorganisms allows for identification of the genetic information for enzymes active on the plastic of interest. The main bottleneck of this approach (function-based) is the availability of efficient and fast screening methods, as recently developed for PET [224,225,226]. Therefore, novel high-throughput screening methods suitable for enzymes and microorganisms must be developed for other polyesters, including polyester-based PURs, as shown by Xu et al. [227].

A different approach called sequence-based screening consists of sequencing environmental DNA and probing only sequences predicted to code for the enzyme(s) responsible for the degradation of the polyester of interest [228]. The advantage of this approach is that only a tiny fraction of the genetic information is experimentally probed, saving time and cost. Moreover, it is noteworthy that this approach can also be applied to DNA sequences already stored in publicly available databases of genomic [229] and metagenomic sequences [230]. Sequence-based screening requires accurate in silico tools with low computational demands to provide a fast, cost-effective, and reliable high-throughput screening protocol to predict whether an enzyme, among several thousand candidates, is able to efficiently degrade the target polyester. An in silico protocol was proposed by Vasina et al. to predict a “small-but-smart” set of efficient biocatalysts for other classes of enzymes [231]. A factor that massively affects the predictive accuracy of these tools is the capacity to disentangle the molecular mechanisms that allow for superior polyester degradation. To achieve this goal, reliable and comparable data on the performance of known enzymes in precise experimental conditions are mandatory.

Therefore, the compilation of a unified, reliable, and user-friendly up-to-date database of the sequence, structure, and functional information of plastic/polyester-degrading enzymes and their organisms would be particularly useful. At present, three resources are available: the Plastics Microbial Biodegradation Database [232], PlasticDB [233], and PAZy [234]. However, information does not seem to be shared and is not uniformly reported, although PAZy was indicated to include only functionally verified enzymes [235]. Nevertheless, recently developed machine learning techniques, which have increased the accuracy of protein structure and function predictions from sequences [236,237], may help in the development of effective sequence-based approaches.

Once good candidate enzymes able to degrade a polyester of interest have been experimentally verified, their activity can be increased by protein engineering [41], i.e., by the introduction of favorable mutations in their sequence. The most used approaches are directed evolution [94], site-directed/saturation mutagenesis [95,238], and machine-learning-based techniques [93,239]. Other common alternatives are the combination of more enzymes in synergy [89], the formation of chimaeras with other proteins that have a hydrophobic surface with polymer-binding properties [240], and the immobilization of enzymes to increase reusability and performance stability over time [241]. Multiple strategies can be achieved simultaneously [242]. Such efforts could increase the enzymatic cleavage rate under higher plastic loads, which is one of the best strategies to lower the environmental impact of enzymatic closed-loop PET recycling (see above).

Engineered enzymes and organisms have also been used to design an open-loop strategy for the conversion of plastic monomers into high-added-value products (Table 5). These systems are still under active development, and the setup of bioreactors for their scale-up has been demonstrated only as a proof of principle; it will be fundamental to understand whether such systems can reach the industrial-scale in the coming years (see the work presented by Andler et al. [43] for a description of the most recent attempts).

Overall, while PET enzymatic recycling requires further enzyme discovery or engineering studies to improve the degradation efficiency (to overcome the *passivation* issue) under high PET load or for less pretreated materials, new enzyme-based technologies are needed to allow for the biotechnological recycling of less recalcitrant polyester-type PURs and other less biodegradable polyesters.

## 5. Conclusions

The information reported in this review underlines that, at present, enzyme-based biotechnological approaches are limited to supporting PET waste circularity and could be potentially extended to other polyester plastics in general, including some less recalcitrant PURs [243]. The enzyme-based processes available for PET degradation are currently less competitive than traditional technologies; their widespread application at the industrial scale will require the design of efficient waste pretreatment processes and PHEs with superior depolymerization rates, especially on high-load and less pretreated plastics. The possibility of producing non-plastic products through open-loop recycling of PET degradation products is very promising, although still under development. For non-PET polyesters, research is still in its infancy and requires more work to understand the best enzymes and the molecular determinants of superior degradation performance.

The situation is challenging in the case of plastics made with more recalcitrant polymers (polyether-type PURs, PAs, polyolefins, PS, and PVC), which account for >90% of unrecycled plastic fractions. Although biotechnology-based degradation approaches exist, they are still inefficient and cannot yet be considered a practical solution. New promising chemical depolymerization technologies seem to represent the only viable treatment option to allow for a subsequent bio-based system for open-loop recycling of degradation products. However, this chemoenzymatic approach must be evaluated in industrial-scale settings. Moreover, only ideal thermoset materials (polyester- and polyether-based PUR foams) have been tested, while the recycling of consumed recalcitrant thermosets remains a main technical issue, and waste-to-energy is still the employed solution, although it is far from being considered “circular”. While reaching thermoset circularity requires the development of new recycling technologies, rejects of collection and sorting systems represent the most critical step in the recycling of thermoplastics. These systems should be optimized in parallel with the introduction of new technologies, including those that make use of microbial enzymes (the focus of this work), for degradation and recycling of difficult-to-separate materials.

In conclusion, the use of enzymatic microbial biotechnologies for plastic degradation and valorization requires a rethink of the whole plastic value chain, as well as the development and marketing of alternative plastic materials (made with non-food biomass derivatives or aliphatic polyesters that have a biodegradability similar or superior to that of PET), the redesign of plastic packaging and products, the improvement of collection and sorting methods, and their tight integration with new, eventually chemo-assisted bio-based systems of degradation and recycling (competitive from an economical point of view and with a lower environmental impact compared to available technologies). This, in turn, will require stronger co-operation between biotechnologists and stakeholders involved with feedstock resources, material production, and waste management to find novel effective industrial technologies and regulations for a transition to a circular (bio)economy in plastic waste management.

## Figures and Tables

**Figure 2 ijms-24-03877-f002:**
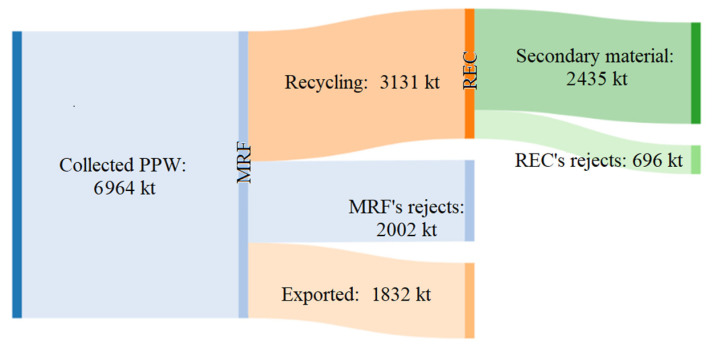
Material flow analysis of EU plastic packaging waste (PPW) in 2017 (kt). The analysis was conducted on five MRFs and eight RECs. Modified from [62]. MRFs: material recycling facilities; RECs: recycling plants.

**Figure 3 ijms-24-03877-f003:**
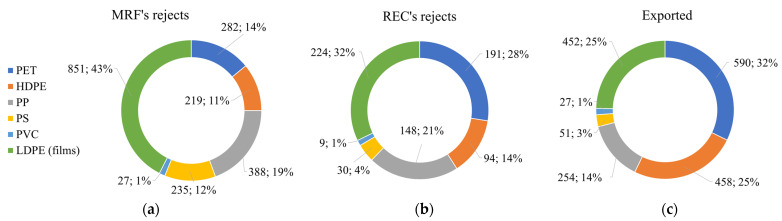
PPW recycling system rejects (2017): (**a**) rejects from MRFs; (**b**) rejects from RECs; (**c**) exported plastic. Values are expressed in kt y^−1^. Adapted from [62].

**Figure 4 ijms-24-03877-f004:**
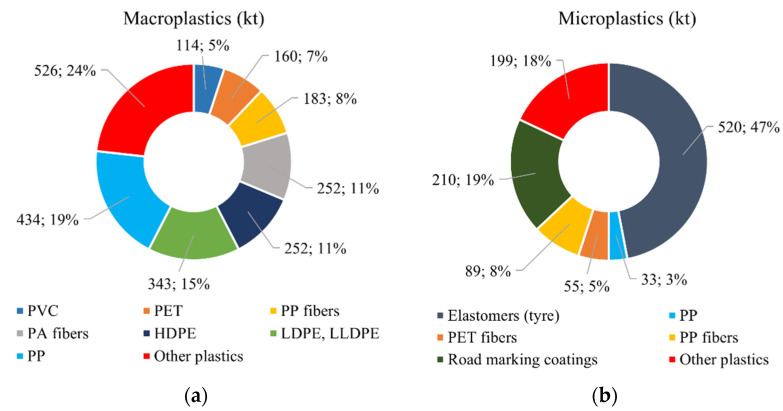
Composition of European plastic inflow into the environment: (**a**) macroplastics; (**b**) microplastics. Elaborated from [51,53]. Absolute amounts expressed in kt.

**Figure 5 ijms-24-03877-f005:**
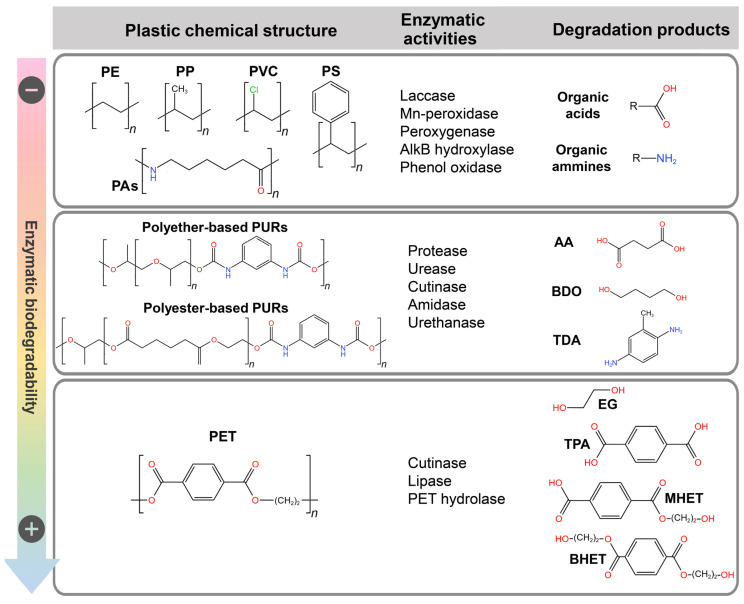
The main classes of enzymes involved in the degradation of different types of traditional petroleum-based plastics in order of their biodegradability to summarize what was reported in the text. Degradation products are exemplified. PURs—polyurethanes, AA—adipic acid, BDO—1,4-butanediol, TDA—toluene diamine, PE—polyethylene, PP—polypropylene, PAs—polyamides, PVC—polyvinylchloride, PS—polystyrene, PET—polyethylene terephthalate, EF—ethylene glycol, TPA—terephthalic acid, MHET—2-hydroxyethyl terephthalate, BHET—bis(2-hydroxyethyl) terephthalate.

**Figure 6 ijms-24-03877-f006:**
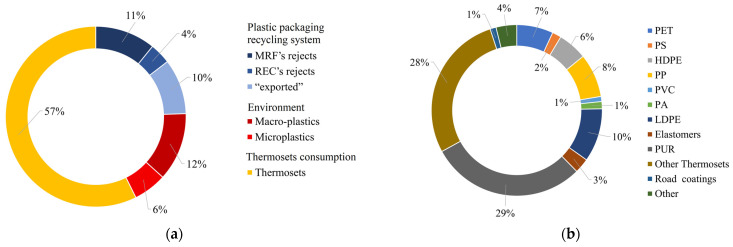
Unrecycled PCPW flows potentially included in future recyclable flows in the EU (Total 18,514 kt y^−1^): (**a**) divided by source; (**b**) divided by polymer composition. Elaborated from data collection and research conducted for this review article.

**Table 1 ijms-24-03877-t001:** List of recent and most relevant general reviews that include biotechnology-based degradation and/or recycling of plastic waste (descending order from the oldest to the newest).

Main Topics	Outlook	Reference
Biotechnological options and challenges for the valorization of plastic waste degradation products with a focus on microbial metabolic engineering.	Few industrially implemented examples until now. The challenges are understanding degradation pathways and measuring the efficiency of microbial conversion to target markets.	[31]
Chemical recycling routes of plastics with reference to life cycle analysis; a small parenthesis on biotechnology; evaluation of industrial systems of sorting and collection.	New or improved chemical catalysts are needed, targeting superior plastic contact and their stability over multiple uses and improving heat distribution. Technological developments will benefit from improvement of cleaning and sorting and from avoiding the commercialization of composite and multilayer materials.	[32]
Catalytic mechanisms and structural rationale of microbial enzymes able to decompose both non-starch plant biomass and synthetic plastics.	Enzymes that can degrade macromolecular polymers (either lignin or recalcitrant plastics) share common biochemical features. Novel plastic polymer-degrading enzymes may be discovered to allow for investigations of the mechanisms by which they operate.	[33]
Biotechnology-based plastic deconstruction; clarification of biodegradable materials and microplastic pollution; challenges and future directions.	Plastics should be made from biomass and CO_2_ and with degradation-on-demand features for products that potentially reach the environment. Consumers must be willing to pay an extra tax to compensate for higher biotech-based recycled plastic prices.	[34]
Microbial degradation of synthetic plastics and probable enzymatic mechanisms.	The biochemical and structural properties of enzymes degrading more recalcitrant plastics need further studies to allow for their modification towards better degradation efficiency. The inclusion of the appropriate pretreatment/additives might yield better results.	[35]
Comprehensive update on challenges and opportunities in chemical and biological catalysis for plastics deconstruction and recycling; suggestions to find standards to compare different mechanisms of plastic deconstruction and their relative performance.	Biological and chemical catalysis should be combined to depolymerize plastics and generate commodity chemicals. These efforts could be synergistic with the development of alternative materials with better end-of-life functionalities that increase their amenability to catalytic deconstruction.	[36]
Enzymatic mechanisms of plastic degradation and factors influencing their performance.	To unravel reaction mechanisms in recalcitrant C-C plastics, basic investigations of changes in substrate polydispersity and the resulting product molecules are required. A ‘bottom-up’ approach of structure-guided de novo enzyme system design is needed.	[37]
Literature survey and challenges of biodegradability of recalcitrant plastics in the presence of pro-oxidants.	There are concerns on microfragments of oxo-plastics reaching marine environments, as no previous study had reported a 100% complete biodegradation of oxo-biodegradable plastic. In the near future, bioplastics are expected to be favored.	[38]
Microbial biodegradation of various synthetic plastic types with a focus on algae and the gut microbial consortium of insects.	The mode of action and mechanism of microbial degradation calls for further studies to detect an effective enzymatic system that fits the tested polymeric material. A practical system for plastic biodegradation is still not available.	[39]
Recent advances in the biotechnology-based biodegradation of both traditional and bio-based plastics with a focus on known degradation mechanisms and valorization of plastic waste.	Studies of the recycling and valorization of plastic waste could offer solutions to plastic industries. Synthetic biology studies on the functioning of microbial cell factories are needed to further improve the adaptability of microbes to a circular economy of plastics; the degradation mechanisms of some types of plastics are still missing and must be studied.	[40]
Comprehensive update on strategies for the discovery and engineering of plastic-degrading enzymes.	New high-throughput screening methods are needed to identify plastic-degrading enzymes. A better connection between protein features and functions is needed to guide accurate protein engineering. The development of engineered thermophilic microorganisms can overcome the problem of short enzyme lifetimes in plastic degradation processes.	[41]
Comprehensive update on plastic depolymerization and upcycling routes, including the modification of plastics to make them more degradable and a mention of biotechnological systems	Academia and industry need to cooperate to create marketable solutions from different plastic recycling technologies with a business model in mind. There should be more international and local government efforts to promote recycling/upcycling and penalize disposal with enforcement. Policies should be introduced to offer more convenient and effective recycling choices for consumers, avoiding low recycling rates due to the collection of highly mixed recyclables. Governments, non-profit organizations, and academia should work together to inform and encourage consumers to choose upcycled and/or biodegradable materials. More interdisciplinary research is needed to create innovative and safe products from plastic upcycling.	[42]
The most promising biotechnological open-loop recycling processes for synthetic plastics with a focus on how to improve degradation with abiotic pretreatments, enzyme engineering, and novel bioreactor designs	Higher degradation activities on polyester and C–C backbone plastics will be fundamental. This can be achieved by engineering existing enzymes and microorganisms, by applying synergistic degradation strategies with multiple enzymes and pretreatments, by focusing on the optimization of the reaction conditions in the reactor, and by evaluating the economic feasibility of plastic monomer upcycling to high-value products	[43]

**Table 2 ijms-24-03877-t002:** Resin types and plastic demand distribution in 2019 (elaborated from [44]).

Resin Type		Acronym	Applications	2019 EU Demand Distribution (Mass %)	Chemical Structure
Thermoplastics	Polypropylene	PP	Food packaging, sweet and snack wrappers, hinged caps, microwave containers, pipes, automotive parts, bank notes, etc.	19.4	–[CH_2_-CH(CH_3_)]_n_–
	Low-Density Polyethylene	LDPE	Reusable bags, trays and containers, agricultural film, food packaging film, etc.	17.4	–(CH_2_-CH_2_)_n_–
	High-Density Polyethylene	HDPE	Toys, milk bottles, shampoo bottles, pipes, houseware, etc.	12.4	–(CH_2_-CH_2_)_n_–
	Polyvinyl Chloride	PVC	Window frames, profiles, floors, wall covering, pipes, cable insulation, garden hoses, inflatable pools, etc.	10	–(CH_2_-CHCl)_n_–
	Polyethylene Terephthalate	PET	Bottles for water, soft drinks, juices, and cleaners; food packaging; textiles; etc.	7.9	–[CO(CH_2_)_4_CO-OCH_2_CH_2_O]_n_–
	Polystyrene	PS + Expanded polystyrene (EPS)	Food packaging (diary, fishery), building insulation, electrical and electronic equipment, inner liner for fridges, eyeglass frames, etc.	6.2	–[CH_2_-CH(C_6_H_5_)]_n_–
	Other Thermoplastics	ABS, PBT, PC, Polytetrafluoroethylene (PTFE), PMMA.	Hub caps (ABS), optical fibers (PBT), eyeglasses lenses, roofing sheets (PC), touch screens (PMMA), cable coating in telecommunications (PTFE), and many other applications in aerospace, as well as medical implants, surgical devices, membranes, valves and seals, protective coatings, etc.	11.3	Many formulations
Thermosets	Polyurethane	Polyurethane (PUR)	Building insulation, pillows, mattresses, insulating foams for fridges, etc.	7.9	–[R-OCO-NH-R2-NH-CO-O]_n_–
	Other Thermosets ^1^	Phenol formaldehyde resins (PF), urea–formaldehyde (UF)	Decorative laminates, textiles, paper, foundry sand molds, foam insulation, paints, coatings, adhesives, etc.	7.5	-

^1^ Includes other thermosets such as phenolic resins, epoxide resins, melamine resins, urea resins, and others.

**Table 3 ijms-24-03877-t003:** Composition of PPW and performance at the different stages of the recycling system. Values are medians and expressed in percentage. Data are taken from [62,63] (PPW collection rate).

Polymer Type	PPW Composition	PPW Collection Rate	MRF Sorting Rate	REC Recycling Rate
PET	18	62	85	81
HDPE	20	44	85	88
PP	20	32	64	66
PS	7	30	37	66
PVC	3	20	73	80
LDPE (films)	32	36	59	71

**Table 4 ijms-24-03877-t004:** List of the most performant biological systems for traditional petroleum-based plastic degradation known to date. They are ordered by plastic polymer type and listed by decreasing degradation rate.

Plastic Polymer	Material	Source	Reaction Condition	Degradation Rate	Reference
PET	Ultrathin PET film (2.5 to 7 nm) obtained from amorphous PET sheets of 2 mm thickness	IsPETaseTM [87] and IsMHETaseSM [88], two variants of *I. sakaiensis* PETase and METase	PBS buffer, pH 7.4, 50 °C	≈70% in 1 h	[89]
PET	Amorphized and micronized PET (≈200–250 µm) from post-consumer bottle-grade PET	Cutinase from leaf–branch compost (ICCG variant)	100 mM potassium phosphate buffer, pH 8.0, 72 °C	≈90% in 10 h	[90]
PET	2 × 1 cm^2^ amorphous Goodfellow PET film and low-crystallinity (13%) PET powder	PHL7/PES-H1, a cutinase from a compost site (L92F/Q94Y variant)	1 M potassium phosphate buffer, pH 8.0, 72 °C, shaking at 1000 rpm	≈100% in 24 h	[91]
PET	3 × 0.5 cm^2^ flakes of an amorphous PET clamshell container	PHL7/PES-H1, a cutinase from a compost site	1 M potassium phosphate buffer, pH 8.0, 70 °C, shaking at 1000 rpm	>95% in 24 h	[92]
PET	Low-crystallinity (1.2 to 6.2%) discs (6 mm) from 51 different post-consumer PET products	FAST-PETase, a variant of *I. sakaiensis* PETase	100 mM potassium phosphate buffer, pH 8.0, 50 °C, shaking at 180 rpm	≈100% in 1 to 7 days	[93]
PET	PET/PE composite packaging tray lid (4 mg, thickness of 325 μm PET and 40 μm PE)	HotPETase, a directed-evolved variant of *I. sakaiensis* PETase	pH 9.2, 50 mM gly-OH buffer with 4% BugBuster	≈20% in 24 h	[94]
PET	≈37% crystallinity PET microplastics (≈300 µm)	TS-ΔIsPET, a variant of PETase from *Ideonella sakaiensis*	100 mM potassium phosphate buffer, pH 8.0, 40 °C, shaking	≈26% in 2 days	[95]
Polyester-type PUR	Lab-prepared 10 µm thin PUR and segmented PUR urea films based on lysine diisocyanate	Papain, Bromelain, Ficin, Chymotrypsin, Proteinase K	PBS, pH 7.0, 37 °C	Up to ≈50% in 7 days	[96]
Polyester-type PUR	Impranil-DLN^®^	*Pseudomonas putida* A12	pH 8.0, 25 °C	92% in 4 days	[97]
Polyester-type PUR	Impranil-DLN^®^	*Pestalotiopsis* *Microspora* E2712A	25 °C in a rotary incubator	99% in 2 weeks	[98]
Polyether-type PUR	PUR foam	*Tenebrio molitor*	Guts of the larvae (probably microbiota-assisted)	67% after 30 days	[99]
Polyester-type PUR	Lab-prepared 0.3 mm thin polycaprolactone thermoplastic PUR film (Capa 2302)	Amidase E4143 and esterase E3576	PBS, pH 7.0, 37 °C	33% in 51 days	[100]
Polyester-type PUR	Impranil-DLN^®^	*Cladosporium pseudocladospo-rioides*, *Cladosporium tenuissimum*, *Cladosporium asperulatum*, *Cladosporium montecillanum*, *Aspergillus* *fumigatus*, *Penicillium chrysogenum*	25 °C, no shaking	40–87% in 2 weeks	[101]
Polyester-type PUR	Commercial 1 mm thin PUR film	*Aspergillus flavus* ITCC 6051	28 °C under 120 rpm shaking	60.6% in 30 days	[102]
Polyester-type PUR	Lab-prepared ~0.2 mm thin PUR	*Bacillus subtilis* MZA-75 and *Pseudomonas aeruginosa* MZA-85	37 °C under 150 rpm shaking	40% in 30 days	[103]
Polyether-type PUR	PUR foam used for commercial production of mattress cushioning	*Cladosporium pseudocladospo-rioides*, *Cladosporium tenuissimum*, *Cladosporium asperulatum*, *Cladosporium montecillanum*, *Aspergillus* *fumigatus*, *Penicillium chrysogenum*	25–30 °C, no shaking	10–65% in 21 days	[101]
Polyether-type PUR	Cubical ether–PUR (1 cm^3^)	*Alternaria* sp. PURDK2	30 °C, no shaking	27.5% in 70 days	[104]
LDPE	LDPE bag	*Galleria mellonella*	-	13% in 14 h	[105,106,107]
LDPE	LDPE film	*Pseudomonas citronellolis*	37 °C under 15 rpm shaking	17.8% in 4 days	[108]
LDPE	LDPE powder	*Cupriavidus necator* H16	30 °C in a rotary incubator	33.7% in 21 days	[109]
LDPE	LDPE foam	*Tenebrio molitor*	Gut of the larvae (probably microbiota-assisted)	49.0% in 32 days	[110]
LDPE	20 μm thin PE film	*Oscillatoria subbrevis*	-	30% in 42 days	[111]
PS	PS pyrolysate oil	*Pseudomonas* *putida* CA-3	Four consecutive treatments at 30 °C for 48 h	10% in 8 days	[112]
PS	Brominated high-impact PS	*Bacillus* sp.	30 °C, no shaking	23.7% in 30 days	[113]
PS	Brominated high-impact PS	*Exiguobacterium* sp. strain YT2	30 °C under 150 rpm shaking	12.4% in 30 days	[114]
PP	Max 250 µm microplastic PP	*Bacillus cereus*	33 °C under 150 rpm shaking	12% in 40 days	[115]
PP	Isotactic PP strips	*Pseudomonas* sp., *Vibrio* sp., *Aspergillus niger*	pH 7.0, 30 °C	60% in 175 days	[116]

## Data Availability

No new data were created or analyzed in this study. Data sharing is not applicable to this article.

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
