# Peer review of "Microbial Enzyme Biotechnology to Reach Plastic Waste Circularity: Current Status, Problems and Perspectives"

_ijms, 2023, doi:10.3390/ijms24043877_

Round 1

Reviewer 1 Report

This paper reviews microbial enzyme engineering techniques for recycling in relation to plastic waste management in Europe. It is appropriate for publication because it deals with the past, present, and future developments of microbial enzyme engineering and is expected to provide useful information to readers. However, there are minor modification that need to be changed for publication. 

Minor :

1. I suggest amending the title to one that more clearly shows the content of the paper rather than its contribution to the plastic waste circularity. In the text, the current researches, limitations (problems), and development direction are discussed, so a more appropriate expression would be better than the “contribution”.

2. Make all three graphs the same size in Figure 3.

Author Response

The authors thank the Referee for the general positive comment. Changes are marked with “Track Changes” function of MS Word.

- The title was changed to “Microbial enzyme biotechnology to reach plastic waste circularity: current status, problems and perspectives” to better underline the content of the paper, as suggested.

- Figures with formatting or text size problems were improved.

Reviewer 2 Report

Dear Authors,

First and foremost I would like to express my congratulations for the hard and precise work you have done on preparing the submitted manuscript "The contribution of microbial enzyme biotechnology to plastic waste circularity" (under ID ijms-2205292). I have carefully read and reviewed the paper you have introduced; please find my comments, observations and recommendations below.

The paper deals with a more-than-ever important subject of environmental science and protection, since the plastic waste load is increasing rapidly globally. The Review article is written in a very detailed and logical way and addresses the most important topics related to the problem, supported by specifically recent references, and self-made figures for better comprehensibility. 

Only some minor corrections and revisions are recommended:

- The use of English and scientific formulation is fine, however some slight improvements would be favorable to correct some typos and other minor grammatical and/or semantical flaws (e.g. line 62 "UE", line 199 "consist of" instead "consists in", line 228 "an [,] is missing after Currently", line 290 should be reformulated (infancy is a bit slangish), line 474 "the use of an [...] was proposed" would sound better) 

- The intertextual citations are correct except for line 266, where "(Ronkay et al., 2021)" is presented instead of the numerical brackets 

- Using the form of expression as "In [n]" or "According to [x]" is not too "elegant" or appropriate towards the original authors. Instead, using "The paper written by XY et al." or "As presented by XY et al." or the like would be more fitting (citations that are concerned e.g. 51, 88, 147, 149, etc.) 

- The sentence line 477-479 covers ("An engineered [...]") is a bit unclear and/or not so easy to comprehend its true meaning; reformulation is advised. 

- The text of the figures that contain "doughnut" charts (e.g. Figure 1, Figure 3) are a bit too small in my point of view in standard (100%) size; if it is possible, a slight enlargement would make them more visible 

- The reference(s) that belong(s) to Figure 5 should be cited right after the text of the Figure 

- Figure 6 is not addressed intertextually 

- At the end of line 301, Authors are sure they wanted to cite Table 1, instead of Table 4? 

- Since Table 4 is especially spacey and covers lots of (really useful) information, my recommendation would be to separate the different types of plastic polymers in a more visible way, for example use bold lines at the end of each polymer variant, or so. 

- DOI numbers should be provided in the Reference list when it is available

Author Response

1)

- The use of English and scientific formulation is fine, however some slight improvements would be favorable to correct some typos and other minor grammatical and/or semantical flaws (e.g. line 62 "UE", line 199 "consist of" instead "consists in", line 228 "an [,] is missing after Currently", line 290 should be reformulated (infancy is a bit slangish), line 474 "the use of an [...] was proposed" would sound better) 

- The intertextual citations are correct except for line 266, where "(Ronkay et al., 2021)" is presented instead of the numerical brackets 

- Using the form of expression as "In [n]" or "According to [x]" is not too "elegant" or appropriate towards the original authors. Instead, using "The paper written by XY et al." or "As presented by XY et al." or the like would be more fitting (citations that are concerned e.g. 51, 88, 147, 149, etc.) 

- The sentence line 477-479 covers ("An engineered [...]") is a bit unclear and/or not so easy to comprehend its true meaning; reformulation is advised. 

1) The authors thank the Referee for the general positive comment and the valuable feedback that was used to improve the submitted manuscript both in terms of language and contents. Changes are marked with “Track Changes” function of MS Word.

2)

- The text of the figures that contain "doughnut" charts (e.g. Figure 1, Figure 3) are a bit too small in my point of view in standard (100%) size; if it is possible, a slight enlargement would make them more visible 

- The reference(s) that belong(s) to Figure 5 should be cited right after the text of the Figure 

- Figure 6 is not addressed intertextually 

- At the end of line 301, Authors are sure they wanted to cite Table 1, instead of Table 4? 

2)

  • Figures with formatting or text size problems were improved.
  • Figure 5 was provided to give a simplified picture to known enzymes and and degradation products, which refer to the detailed explanation presented and cited in the text.
  • Figure 6 is now cited intertextually
  • The citation to Table 1 is what authors wanted, as it comes at the end of a general statement on plastic biodegradation attempts, and the Table 1 reports the previous reviewing works on such attempts

3)

- Since Table 4 is especially spacey and covers lots of (really useful) information, my recommendation would be to separate the different types of plastic polymers in a more visible way, for example use bold lines at the end of each polymer variant, or so.

3)

- Table 4 was reformatted to follow the suggestions.

4)

- DOI numbers should be provided in the Reference list when it is available

4)

- DOI numbers were added when available

Reviewer 3 Report

The manuscript by Orlando et al reviews the state of the art on microbial enzymes for plastic waste recycling. 
The authors described and summarized the literature in a very clear manner to permit a better understand of the missing points. 

The manuscript is well written and the reviewer has only few comments which could enhance the manuscript value. 

Fig. 1. The authors describe the different applications of plastics but for the recycling they focus mostly on packaging. There is an interesting literature worth mentioning also on textile recycling such as: 
Quartinello et al 2017 Microbial Biotechnology 10, 6, 1376-1383
Jönsson et al 2021 ChemSusChem 14, 19, 4028-4040

Table 4: The reviewer suggests to list the polymers by type first which would help the reader to focus on the search 

In general and in particular on Fig. 5, authors do not describe polymers containing aliphatic moieties such as PBAT, PLA and PCL. This could increase the interest of the review for the scientific community. 

The authors describe variants of wild-type enzymes but only for the very well-known IsPET. Other variants of enzymes such as lipases, cutinases or proteases are not mentioned in the review. The reviewer suggests to increase the scope on different substrates, like polyamides and nylons, and to include variants with different specificity, respectively:
Negoro S 2000 Applied and Microbial Technology 54, 4, 461-466
Biundo et al 2019 RSC Advances 62, 9, 36217-36226

Author Response

The authors thank the Referee for their general positive comments and the valuable feedback that was used to improve the submitted manuscript especially in terms of contents. Changes are marked with “Track Changes” function of MS Word.

1) Fig. 1. The authors describe the different applications of plastics but for the recycling they focus mostly on packaging. There is an interesting literature worth mentioning also on textile recycling such as: 
Quartinello et al 2017 Microbial Biotechnology 10, 6, 1376-1383
Jönsson et al 2021 ChemSusChem 14, 19, 4028-4040

1) The manuscript was modified in different points (Lines 245-247; Lines 431-436; Lines 468-470) to underline the problem of textile recycling due to blended fibers, and the suggested works were added as references.

2) Table 4: The reviewer suggests to list the polymers by type first which would help the reader to focus on the search

2) Table 4 was reformatted to follow the suggestions.

3) In general and in particular on Fig. 5, authors do not describe polymers containing aliphatic moieties such as PBAT, PLA and PCL. This could increase the interest of the review for the scientific community. 

The authors describe variants of wild-type enzymes but only for the very well-known IsPET. Other variants of enzymes such as lipases, cutinases or proteases are not mentioned in the review. The reviewer suggests to increase the scope on different substrates, like polyamides and nylons, and to include variants with different specificity, respectively:
Negoro S 2000 Applied and Microbial Technology 54, 4, 461-466
Biundo et al 2019 RSC Advances 62, 9, 36217-36226

3) The authors agree to these comments since this information could enlarge the overview of “potentially enzyme-treatable” polymers. The suggested references, indicated by referee 3, were used to extend the description of biodegradative systems active on nylon oligomers (Lines 627-629). The authors are aware that there are several promiscuous cutinases, esterases and lipases that may (potentially) be suitable to degrade not just PET, but also other types of plastics (PAs, PCL, PLA, PBAT, etc.), but exhaustive treatment of these polymers (e.g., indication of improved engineered variants present in literature, especially for more recalcitrant, PAs such as nylons, or less commercialized polymers, such as PCL, PLA, PBAT) is beyond the focus of the current review which is centered on the plastic types that mostly contribute to the actual waste cycle from a quantitative point of view and for which reliable LCA assessments are available. In addition, from a biochemical point of view, this review is focused on systems that are most promising to date (i.e., with a superior degradation rate) for treating the plastic polymers. While promising results were obtained on PET circularity from biodegradative processes, much less is known about other types of polymers, for which the investigation of degradative enzymes is yet at the very beginning. For this reason, the description of less studied degradative systems for other polyester-based polymers (PCL, PLA, PBAT, etc.) will be considered in a future review focused on emerging polymers (bio or not).